# A prospective pilot study assessing levels of preoperative physical activity and postoperative neurocognitive disorder among patients undergoing elective coronary artery bypass graft surgery

**Setayesh R. Tasbihgou**[1]*, **Sandra Dijkstra**[2], **Sawal D. Atmosoerodjo**[1], **Iris Tigchelaar**[3], **Rolf Huet**[1], **Massimo A. Mariani**[2], **Anthony R. Absalom**[1]

**1** Department of Anesthesiology, University Medical Centre Groningen, University of Groningen, Groningen, The Netherlands, **2** Department of Cardiothoracic Surgery, University Medical Centre Groningen, University of Groningen, Groningen, The Netherlands, **3** Laboratory for Experimental Ophthalmology, University Medical Centre Groningen, Groningen, The Netherlands

* s.r.tasbihgou@umcg.nl

## Abstract

Physical inactivity and a sedentary lifestyle are associated with a chronic low-level inflammatory state which has been implicated in the pathogenesis of cardiovascular disease. There is growing interest in exercise programs as part of surgical 'prehabilitation'. We therefore studied preoperative physical activity levels of patients undergoing elective Coronary Artery Bypass Graft (CABG) surgery, and performed an exploratory analysis of the influence of physical activity on postoperative outcome. The Short Questionnaire to Assess Health (SQUASH) was used to assess physical activity among 100 patients, of mean (SD) age 65.4 (7.6) years. Additionally, handgrip strength was measured, and the get-up-and-go test was conducted. Anxiety, depression, and quality of life were assessed, and a computerised cognitive test battery was used to assess cognitive performance preoperatively, and three months after surgery. Preoperatively, 76% of patients met the recommended national guidelines for physical activity. The incidence of pre-existing medical conditions, and other preoperative patient features were similar in active and inactive patients. Preoperative physical activity was significantly inversely related to the logistic EuroSCORE. The level of physical activity was also significantly inversely related with preoperative C-reactive protein (CRP) and peak postoperative CRP, but physical activity did not appear to be associated with any adverse postoperative outcomes or extended length of hospital stay. The incidence of postoperative neurocognitive disorder (PNCD) at 3 months postoperatively was 26%. Cognitive performance was not related with physical activity levels. In summary, this was the first study to assess activity levels of cardiac surgical patients with the SQUASH questionnaire. The majority of patients were physically active. Although physical activity was associated with lower levels of inflammation in this pilot study, it was not associated with an improved clinical or cognitive postoperative outcome.

**Data Availability Statement:** All relevant data are within the manuscript and its Supporting Information files.

**Funding:** The authors have received no specific funding for this work

**Competing interests:** SRT, SWD, IT and RH have declared that no competing interests exist. SD has read the journal's policy and the authors of this manuscript have the following competing interests: grants from Stichting Beatrixoord Noord-Nederland during the conduct of the study. This grant is not related in anyway to the study; MAM has read the journal's policy and the authors of this manuscript have the following competing interests:consultancy from AtriCure, Getinge and LivaNova; ARA has read the journal's policy and the authors of this manuscript have the following competing interests: reports unrestricted research and/or consultancy for The Medicines Company, Janssen Pharma, Carefusion/BD, Orion Pharma, Ever Pharma, and Philips (all for work unrelated to the current study; all payments to institution); and being Editor of the British Journal of Anaesthesia. This does not alter our adherence to PLOS ONE policies on sharing data and materials.

## Introduction

In recent years, there has been considerable interest in the relationship between physical activity, fitness, and health. For patients undergoing surgery, not only does pre-operative physical activity have the potential to improve their general health, it also might influence the outcome of the operation itself [1–4]. For cardiac surgery however, physical activity is a double-edged sword. Although a sedentary lifestyle (i.e. physical inactivity) is a known risk factor for cardio-vascular disease [5], patients may be unwilling, or even unable to comply with exercise programs because of fear, or because their exercise tolerance is limited by angina or dyspnoea.

The disadvantages of a sedentary lifestyle are well-known, and the underlying mechanisms have also been determined [6]. Physical inactivity has been shown to be associated with a chronic-low level inflammatory state [6–8]. This systemic inflammatory state plays a key factor in the pathogenesis of diseases such as atherosclerosis and insulin resistance [9, 10]. Indeed, an inactivity-associated inflammatory state has consistently been associated with a decreased health related quality of life (HRQL), and an increased incidence of non-communicable diseases, such as coronary heart disease (CHD) and type II diabetes mellitus [8, 11–13]. It has also been associated with colon- and breast cancer [8, 11].

There is growing evidence that physical activity is likely associated with improved cognitive function, probably mediated by an attenuation of brain inflammation [6, 14–17]. A recent study comparing physically active and inactive persons showed indeed evidence of this relationship: that inadequate levels of physical activity were associated with higher levels of pro-inflammatory marker IL-12p40, smaller lateral prefrontal cortical and hippocampal volumes, and worse cognitive function over a 6 year period [18]. These findings were consistent with those of previous studies showing that higher levels of pro-inflammatory markers such IL-6 are associated with reduced overall grey matter and hippocampal volume, and worse cognitive performance [19–22].

Frailty is an age-related decline in multiple physiological systems which renders affected individuals susceptible to sudden health deterioration in response to stressors [23]. Among cardiac surgical patients, frailty has been associated with increased length of hospital stay, mortality, major adverse cardiovascular and cerebral events, and a reduced functional status [24–26]. It is also associated with neurocognitive complications such as delirium and postoperative neurocognitive disorder (PNCD) [27, 28]. Given the above-mentioned associations and the widely held perception that patients with ischemic heart disease mostly have sedentary lifestyles, the interest in preoperative rehabilitation (prehabilitation) has also been focused on patients undergoing cardiac surgery. However, a prehabilitation program, often consisting of preoperative physical and/or cognitive therapy, would probably be more beneficial for physically inactive patients awaiting cardiac surgery.

The main aim of the current study was thus to generate an inventory of preoperative levels of physical activity, and compare these to the recommendations of the Dutch National Institute for Public Health and Environment (Rijksinstituut voor Volksgezondheid en Milieu, RIVM), among patients undergoing coronary artery bypass graft surgery (with or without the use of cardiopulmonary bypass) in our hospital. The secondary goals were to assess the feasibility and performance on tests indicating sarcopenia and frailty (grip strength, and the 'get-up-and-go' test), quality of life questionnaires in cardiac patients undergoing surgery. Given the purportedly high incidence of cognitive decline among elderly patients after cardiac surgery [29–31], and the potential associations among physical activity, inflammation, and cognitive function (and the possible role of inflammation in PNCD [26, 32]) we also assessed cognitive function before and after surgery.

## Materials and methods

After approval of the study by the Groningen Medical Ethics Committee (approval reference number: NL49262.042.14 [METc 2014/219 Sarcopenie]), 100 patients undergoing elective coronary artery surgery were enrolled in this prospective longitudinal observational study performed at the University Medical Centre Groningen (UMCG), the Netherlands, between January 2015 and March 2017. All subjects completed and signed an informed consent form prior to their enrolment in the study.

Inclusion criteria were age > 18 years old and suitability for undergoing coronary artery surgery with or without cardiopulmonary bypass (CPB). Patients operated for coronary artery disease combined with a valve procedure and reoperations will not be eligible for inclusion. Patients were further required to be able to stand and walk independently, and to be prepared to allow a researcher to repeat the assessments three months after surgery, at their own home or at the hospital. Patients are required to participate in online screening modules for cognitive function (ie they should be able to operate a computer touch pad or mouse, and to read large text on a computer screen). Furthermore, patients need to be able to perform the handgrip strength test on both sides.

Patients were also excluded from participation if they were unable to understand or read Dutch instructions. Patients expected to have an extended postoperative intensive care stay, or those with a diagnosed history of a recent depression, anxiety, dementia, stroke or other neurological disorders or severe cerebrovascular insults were also excluded from participation. Finally, patients not able to perform the get-up-and-go test or any of the other tests were also considered ineligible for this study.

### Primary outcome measurements

Data collection was conducted at four time points; at the preoperative anaesthetic screening visit (4–6 weeks before surgery), 1 day before surgery, 4 days after surgery and finally 3 months after surgery (Fig 1). All questionnaires, assessments, and neuropsychological tests were administered by trained research assistants. The surgery and anesthesia for all patients were performed in a conventional manner with routine pre and postoperative care.

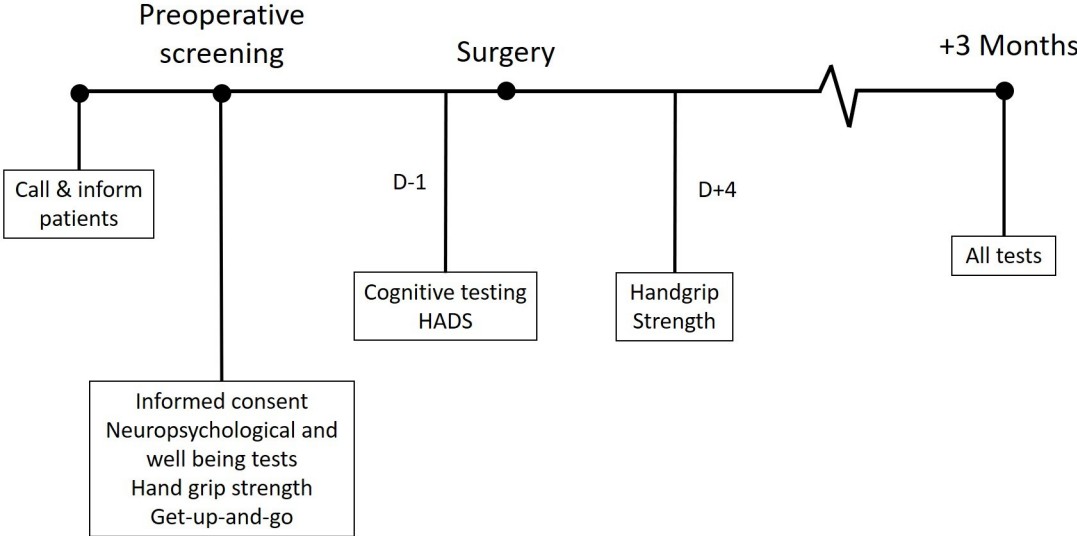

**Fig 1. Data collection timeline.** A timeline of data collection per patient. D-1 = one day before surgery; D+4 = four days after surgery; HADS = Hospital Anxiety and Depression Scale.

**Physical activity.** The primary outcome of this study was perioperative levels of physical activity, assessed using the Short Questionnaire to Assess Health (SQUASH)(24), at the preoperative screening visit and 3 months after surgery. The SQUASH questionnaire was developed in the Netherlands, and has been validated using an activity monitor in both the healthy adult population and in patients who have undergone total hip arthroplasty [33, 34]. It has been adopted by various Dutch governmental health-institutions to assess fitness and health activity in the general population. The 2011 SQUASH questionnaire modified by the Dutch Centre for Big Data and Statistics (CBS) was selected for this study. The questionnaire uses a combination of open and closed questions to determine the amount of time and effort spent on various activities (S1 Table). Using the Ainsworth compendium of physical activities, each activity is assigned a Metabolic Equivalent of Task (MET) value (S2 Table) [35, 36]. The amount and intensity of these activities can then be allocated to one of three categories defined by the RIVM at the time of the study; Dutch Norm of Healthy Activity (Nederlandse Norm Gezond Bewegen—NNGB), Fitnorm and Combinorm (Table 1) [37–39].

For the purposes of the current study, we defined a patient as being physically inactive if they fulfilled the Combinorm criteria of inactivity: less than 30 minutes of moderate activity per day for 5 days a week <u>or</u> less than 20 minutes of intense activity for 3 days a week. Postoperative statistical analyses were also performed using '*Activity scores per week'* from the SQUASH [34]. In order to calculate the *activity score*, first an *intensity score* is assigned to an activity based on the MET-value for that activity and the degree of effort the patient's report doing these activities (Table 2). The *activity score* is calculated by multiplying the total amount of minutes spent on an activity by its *intensity score*. Activity scores are then calculated for light, moderate and vigorous activities. The *total activity score* per week is then calculated taking the sum of all the activity scores for light, moderate and vigorous activities.

After the study started, the ethical committee approved an amendment to allow an objective measurement of movement and energy expenditure, using the portable Sensewear apparatus (Body Media, Pittsburgh, PA, USA). These data will be reported separately (manuscript by Dijkstra et al. is yet to be submitted).

## Secondary outcome measurements

**Handgrip strength and 'get-up-and -go' test.** Further measurements were used to assess muscle strength and mobility: handgrip strength and the get-up-and-go test respectively. Handgrip strength was tested bimanually using a JAMAR™ hydraulic hand dynamometer (Sammons Preston Ltd, Bolingbrook, IL., USA). The subjects performed the test three times per hand with the elbow at the side of their body at 90 degrees flexion and the wrist rotated in neutral position with thumbs towards the ceiling, for maximum isometric force according to

**Table 1. Dutch National Institute for Public Health and Environment (RIVM) recommendations for duration and intensity of physical activity.**

| | Time | Intensity | |
|---|---|---|---|
| | | Adults (18–54 yrs) | Elderly (55+ yrs) |
| **NNGB** | At least:<br>• 30 minutes a day, 5 days a week | Moderate intensive activities at 4–6.5 MET | Moderate intensive activities at 3–5 MET |
| **Fitnorm** | At least:<br>• 20 minutes a day, 3 days a week | Vigorous intensity activities at ≥6.5 MET | Vigorous intensity activities at ≥5 MET |
| **Combinorm** | Activity levels sufficient if at least one of the NNGB and Fitnorm criteria are met. | | |

NNGB, Nederlandse Norm Gezond Bewegen; MET, Metabolic Equivalent of Task.

**Table 2. Intensity scores assigned to various levels of activity and self-reportred effort.**

| Ainsworth and Dutch norms for the intensity of an activity | Self-reported intensity/effort level | | |
|---|---|---|---|
| | Light | Moderate | Vigorous |
| Light | 1 | 2 | 3 |
| Moderate | 4 | 5 | 6 |
| Vigorous | 7 | 8 | 9 |

the instrumental protocol. Handgrip strength was expressed in kilograms. The mean of these measurements was calculated per hand. This mean value was then expressed as a percentage value of age and gender matched control data [40]. Impairment of handgrip strength was indicated when the strength of the dominant hand was below <85% [41].

The Get-up-and-Go test entails the time taken for a patient to get up from a seated position (a chair without side-supports), walk 3 meters, and return to their seat at their normal pace. This test was performed twice, and the mean time was selected for further analysis. Frailty is indicated in patients who take longer than 12 seconds to complete the task [42].

**Cognitive function testing.** Cognitive function was assessed by administering the Cog-State brief computerised cognitive test battery (Cogstate Ltd, Melbourne, Vic., Australia). The test-battery was conducted twice before surgery, the first as a practice test and the second as a baseline measurement [43]. The tests were repeated 3 months after surgery. The test battery consists of four tests; a detection task, an identification task, a one card learning task, and the one back task. These tests assess four neurocognitive domains; psychomotor speed, selective attention, long-term memory and working memory, respectively [44]. Impairment within these domains is typical for postoperative neurocognitive disorder [45]. Each round of testing took approximately 15–20 minutes. Standardised written instructions on how to perform the tasks were included in the test battery in order to minimise inter-observer variability. These test scores were analysed to determine the incidence of PNCD at 3 months after surgery. PNCD was defined as a decline in cognitive performance beyond natural variation. Our approach to analysing the cognitive tests was similar to that outlined by Rasmussen et al. [46]. A standardised change Z-score was calculated for each cognitive test, equivalent to the Reliable Change Index, taking into account the test-retest variability among an age-matched normal control population [46]. These scores were then also summed to generate a composite-Z score. We defined PNCD as present in an individual when their standardised change-Z score was less than the -2 in two or more cognitive domains and/or when their composite-Z score was ≤-2 [46, 47].

**Neuropsychological and wellbeing assessments.** Patients diagnosed with an anxiety or depression disorder were excluded from participation in the study. However, as anxiety and depression are associated with impaired cognitive function [46], we thus screened enrolled patients for pre- and postoperative depression and anxiety using the Hospital Anxiety and Depression Scale (HADS) [48]. Patients completed the questionnaire at the time of preoperative screening, after admission just prior to the start of the first cognitive test session and at 3 months after surgery. A cut-off HADS score of ≥8 was used to identify patients with (undiagnosed) depression and/or anxiety [48, 49]. All neuropsychological tests were administered by trained research assistants.

Questionnaires were used to assess health status, quality of life and health limitations. General health status was assessed using the EuroQol-5D (EQ-5D) survey [50], health-related quality of life perception was assessed using the Research and Development-36 (RAND-36) questionnaire [51], and after another amendment the ethical review board approved the

inclusion of an assessment for health limitations (disabilities) in various domains of functioning using the World Health Organisation, Disability Assessment Schedule (WHODAS-2.0) questionnaire [52]. These assessments were also made at the time of preoperative screening, and 3 months after surgery.

**Other secondary outcomes.** Other secondary outcomes include the routinely collected laboratory values (e.g. C-reactive protein (CRP), leukocytes, and thrombocytes), information regarding medical history, logistic EuroSCORE (a measure of cardiac risk [53]) and the incidence of routinely collected outcome parameters. The latter includes process variables (such as duration post-operative ventilation, ICU stay, hospital stay) and complications (wound infection or breakdown, re-operation, delirium, atrial fibrillation, renal failure).

## Sample size analysis

In a previous study on fitness in healthy male and female septuagenarians, 23% were found to be unfit and 76% were physically active or to have exercised frequently [54]. Although elective CABG patients are expected to be younger, they are also expected to have lower levels of physical activity. It was thus estimated that 50% of the patients who would be included in the study would have an inactive lifestyle. The sample size of 100 was chosen for practical and pragmatic reasons. If, with a sample size of 100, the proportion of physically active patients turned out to be 50%, then the 95% confidence interval (CI) for the true population value of this proportion would be 40.4–59.6%. As such, this study is a baseline measurement or a pilot study, with the sample size pragmatically chosen, and its goal was to inform future studies.

## Statistical analysis

Analyses were performed using SPSS statistics software version 23 (IBM, New York, NY, USA). The normality of distribution of the data was tested by assessing histograms and when appropriate with the Kolmogorov-Smirnov test. Non-parametric data are presented with medians and interquartile range (IQR), whereas, parametrically distributed data are presented with their mean and standard deviation (SD) and binary variables are presented with the incidence and percentage of sample size. Analyses of normally distributed continuous data were conducted using the Student t-test. For non-parametric continuous data, the Mann-Whitney-U test was conducted. For nominal variables, such as gender and the incidence of anxiety or depression, a chi-square test was performed when appropriate. A theory driven approach was used for the selection of variables to be included into correlation and regression analyses. Correlations between variables were analysed using the Pearson or Spearman rank-order correlation coefficient. In the event of a significant correlation, (multiple) linear regression would be used to asses this relationship. To evaluate the potential independent prognostic effect of physical activity on inflammation, a multivariable linear regression analyses was used to control for age and BMI, which are variables known to be associated with inflammation [6, 9, 10]. Univariate and Multivariate logistical regression was used to assess the possible risk factors for physical inactivity and PNCD. If determinants were found significant (p<0.2) in univariate analyses, multivariate regression was performed. Odds ratios (ORs) and their corresponding 95% confidence intervals (95% CIs) were calculated and reported. A two-sided probability value of < 0.05 was considered statistically significant.

## Pre- and post- rehabilitation

After the start of the present study, the department of cardiothoracic surgery introduced a pre- and postoperative rehabilitation program to their standard care package. A number of patients who had already been included into current study were offered the chance to participate in

this program, and eventually 31 patients participated in the prehabilitation program **after** they had completed the SQUASH questionnaire. In addition, 3 patients participated in an alternative in- and outpatient postoperative rehabilitation program.

The prehabilitation program included multiple physical therapy sessions consisting of inspiratory muscle training, aerobic cycling, resistance training and body awareness therapy. Furthermore, multiple sessions of group education, work and psychological guidance, dietary advice and additional counselling in order to quit smoking was also administered. Patients participated in these activities from at least three weeks prior to the surgery. In addition to the group education, psychological guidance and counselling sessions, the alternative postoperative rehabilitation program also consisted of aerobic cycling, resistance training, swimming, and sport and games.

Since participation in these programs could have influenced the secondary outcomes, only the postoperative data of patients who **did not** attend the prehabilitation program will be included in the statistical analyses, using the total activity scores as an indicator for physical activity instead of the dichotomized Combinorm as independent variable. When assessing cognitive function 3 months after surgery, both the patients who attended the prehabilitation program and the alternative postoperative rehabilitation program were excluded from analyses.

## Results

A total of 100 patients undergoing elective CABG were enrolled in the study (see CONSORT diagram, Fig 2). The baseline demographic, clinical and surgical characteristics, of the patients are presented in Table 3. The mean (SD) age of the patients was 65.4 (7.6) years old, and 89%

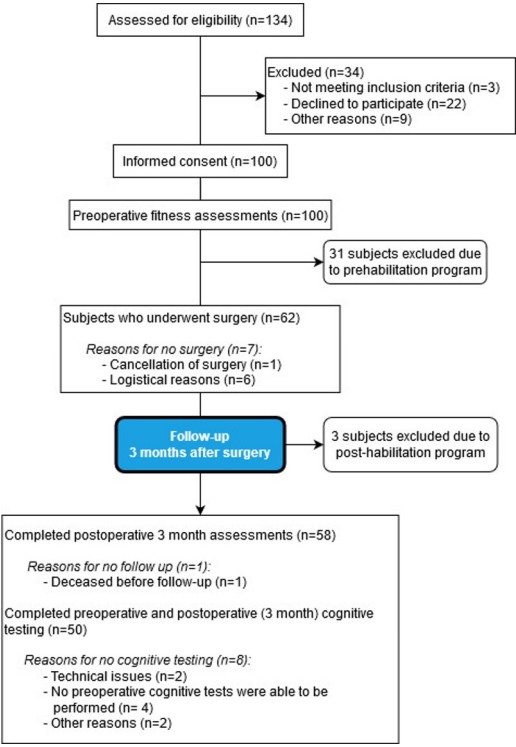

**Fig 2. CONSORT flow diagram.** A consort flow diagram of all the patients included into the study.

**Table 3. Demographic and clinical characteristics of all patients included into the study.**

| | All patients (n = 100) | Preoperatively active (n = 76) | Preoperatively inactive (n = 24) | p value |
|---|---|---|---|---|
| **Demographics** | | | | |
| Male Sex | 89(89%) | 68 (89.5%) | 21 (87.5%) | 0.788 |
| Age, mean (SD), years | 65.4(7.6) | 65.9(7.5) | 63.8(7.9) | 0.238 |
| BMI, mean (SD) | 28.1(4.7) | 27.1 (4.7) | 28.6(4.6) | 0.359 |
| Level of education, median (IQR) † | 5 (4–6) | 5 (4–6) | 5 (4–6) | 0.538 |
| Logistic EuroSCORE, median (IQR) | 2.5(1.4–4.0) | 2.4 (1.4–4.0) | 2.5(1.3–4.4) | 0.681 |
| **Pre-existing medical conditions** | | | | |
| Diabetes | 31(31%) | 22(29%) | 9(38%) | 0.430 |
| Hypertension | 54(54%) | 43(57%) | 11(46%) | 0.357 |
| COPD | 11(11%) | 9(12%) | 2(8%) | 0.632 |
| Respiratory | 12(12%) | 9(12%) | 3(13%) | 0.931 |
| Other | 64(64%) | 47(61.8%) | 17(71%) | 0.424 |
| **Preoperative lab test results** | | | | |
| CRP, median (IQR), mg.l$^{-1}$ | 5(1–4) | 1.8(0.7–3.3) | 3.6(1.5–12) | **0.006** |
| Leukocytes, median (IQR), $\times 10^9$.l$^{-1}$ | 7.7(6.6–8.7) | 7.3(6.7–8.7) | 7.2(6.5–8.9) | 0.831 |
| Thrombocytes, median (IQR), $\times 10^9$.l$^{-}$ | 243(203–277) | 238(203–287) | 242(218–273) | 0.687 |

*Data are presented as number (%), unless otherwise indicated.

† Level of education according to Verhage Classification of Dutch Education Levels, ranging from less than elementary school (1) to university degree (7). BMI = body mass index; COPD = chronic obstructive pulmonary disease; CRP = C-reactive protein; HADS = hospital anxiety and depression score; IQR = interquartile range; SD = standard deviation

of the patients were men. The median waiting time between preoperative anaesthetic screening and surgery was 29 days. The shortest waiting time was 6 days and longest 110 days.

## Pre-operative physical activity

SQUASH questionnaires were completed by all patients. Of the 100 patients, a total of 76 patients fulfilled the Combinorm criteria for a healthy level of physical activity. Fig 3 illustrates

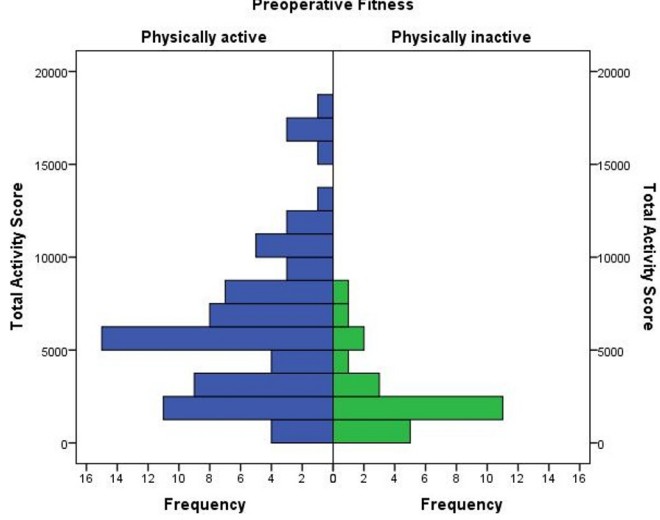

**Fig 3. Physical activity and total activity scores.** A mirrored bar chart of the various total activity scores for physically active (blue) and inactive (green) patients (n = 100).

the total activity scores for physically active and inactive patients. There were no significant differences in any demographic or clinical characteristics, or incidences of pre-existing medical conditions between the two groups. Using univariate and multivariate logistic regression no significant risk factors for physical inactivity were identified (S3 Table). Furthermore, neither age or BMI were significantly correlated with total activity score. There was a statistically significant negative relationship between total activity score and logistic EuroSCORE (Spearman's rho = -.254, p = 0.047).

## Strength and mobility assessments

Preoperative handgrip strength tests were performed in 97 of the 100 patients and again, 4 days after surgery, in 57 (91%) of 62 patients who underwent surgery. Grip strength was similar between physically active and inactive patients (S4 Table). Fifteen patients were identified with an impaired handgrip strength (<85%) preoperatively, of whom 10 (13.2%) were physically active, and 5 (20.8%) were inactive (p = 0.359). Timed get-up and go tests were performed in 96 patients. Of them, five (5.2%) patients took longer than 12 seconds to complete the test. Of these five patients, three (4.1%) were classified as physically active and two (8.7%) as physically inactive according to the Combinorm criterion.

## Cognitive function tests

There were 50 patients who performed all the cognitive function tests and did not participate in the prehabilitation program, and their data were subjected to PNCD analysis. Among these patients, postoperative neurocognitive disorder was identified in 13 (26%) patients at 3 months. All of the patients that fulfilled the criteria for PNCD did so on the basis of a composite-Z score $\leq$-2. Three patients fulfilled the criteria of a standardised change-Z score <-2 in

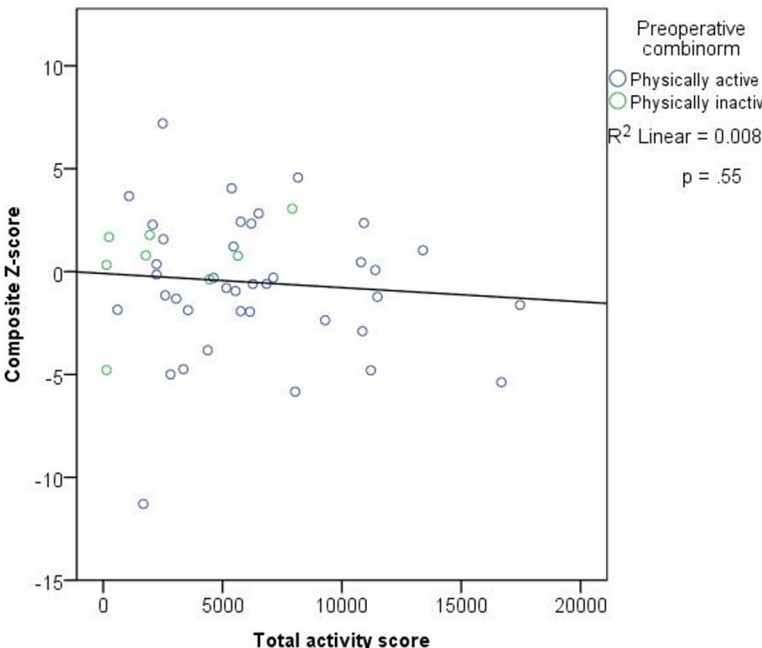

**Fig 4. Total activity scores and composite Z-score.** A linear regression model for total activity score and composite Z-score. There was no significant relationship p = .55. Active (blue) and inactive (green) patients have also been distinguished (n = 47).

≥2 cognitive domains. There was no correlation between the total activity score and the composite Z-score 3 months of surgery (Pearson's rho = -.089, p = 0.550, Fig 4). No other variables were significantly related to composite Z-score. In addition, the composite-Z scores were not associated with preoperative handgrip strength (Dominant hand: Pearson's rho = 0.156, p = 0.279). Through simple linear regression and multiple linear regression no significant predictors were found for composite Z-score. Through logistic regression no significant risk factors (physical activity, impaired handgrip strength, education level, age, the use of CPB, postoperative complications, postoperative anxiety or depression) could be identified for PNCD (S5 Table).

When comparing the baseline cognitive scores for active and inactive patients, no statistically significant differences were identified. The baseline standardised scores for each cognitive domain, for both active and inactive patients, are presented in S6 Table.

## Neuropsychological and wellbeing assessments

The results of these assessments are summarised in S7 Table. During preoperative screening, 14 (18.9%) physically active and 7 (28%) inactive patients had HADS score ≥8 for either anxiety or depression. Anxiety and depression scores were not correlated with the total activity score (Spearmans rho = .053, p = 0.607 and Spearmans rho = -.179, p = 0.081 respectively). Preoperatively physically inactive patients scored significantly lower in 5 domains of the RAND-36; physical function (p = 0.011), social function (p = 0.015), physical role limitations (p = 0.007), emotional role limitations (p<0.001), and vitality (p = 0.014). Nevertheless, no differences between active and inactive patients were measured with the EQ5D and WHODAS. After 3 months, no differences were seen in any of the neuropsychological assessments between the active and inactive patients.

## Other outcomes

Median (IQR) length of stay in the ICU was 1 (1–1) day for both physically active and inactive patients. The median (IQR) length of stay in the hospital after surgery was 7 (6–8) days for the active group and 6 (5–10) days for the inactive group. There were no significant differences in total activity scores between patients with and without postoperative complications.

Preoperative laboratory values for leukocytes and thrombocytes were similar between active and inactive patients. Preoperative median (IQR) CRP levels were 1.8 (0.7–3.3) and 3.6 (1.5–12) for active and inactive patients, respectively. This difference is statistically significant (p = 0.006), but not clinically relevant. The total activity score was also significantly negatively related to preoperative CRP (Spearman's rho = -.259, p = 0.010, n = 100).

Peak postoperative CRP was also correlated with the total activity scores (Pearson's rho = -.346, p = 0.007, n = 59). A simple linear regression was used to predict peak CRP based on preoperative total activity scores (Fig 5). A significant regression equation was found (F (1, 57) = 7.747, p = 0.007), with an $R^2$ of .120. Patients predicted CRP is equal to -0.006 [total activity score] + 197.4. The model for CRP appears to be more accurate for people with lower activity scores. A multiple linear regression was calculated to predict peak postoperative CRP based total activity scores, corrected f or age, and BMI (Table 4). A significant regression equation was found (F(3,55) = 4.566, p = 0.006), with an $R^2$ of 0.199. CRP remained a significant predictor when correcting for age and BMI.

## Discussion

When this study was conducted, the Dutch National Institute for Public Health (RIVM) recommended that people perform ≥ 30 minutes of moderate exercise 5 days per week (the so-called

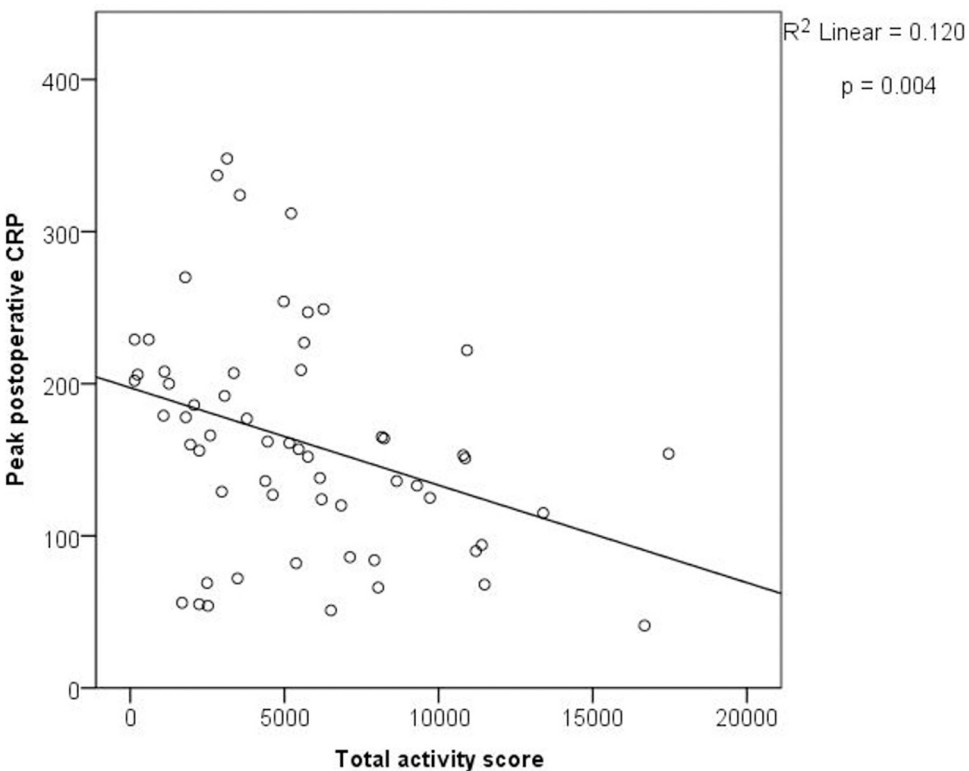

**Fig 5. Total activity score and peak postoperative CRP.** A significant linear regression model between total activity score and peak postoperative C-reactive protein (CRP) (n = 59). Total activity was inversely related to peak CRP (p = .007).

NNGB criteria) **OR** $\geq$20 minutes of vigorous exercise 3 days per week (the so-called Fitnorm). We have shown that during the conduct of our study, 76% of the included patients met at least one of the two criteria (the so-called Combinorm). Preoperatively, patients with lower activity scores were more likely to have higher Logistic EuroSCOREs, suggesting that cardiac morbidity may have a significant adverse effect on physical function (or possibly the converse). Preoperatively, physically inactive patients also reported lower quality of life scores, however this was limited to only some items of the RAND36 and not reflected in the EQ5D or WHODAS.

No potential risk factors for physical inactivity were found. Nevertheless, it is important to recognise that the level of physical activity a person is able to do is multifactorial, determined

**Table 4. Simple and multiple linear regression models for peak postoperative C-reactive protein (n = 59).**

|  |  | B | beta | p value |
|---|---|---|---|---|
| **Step 1** |  |  |  |  |
|  | Total activity score | -0.006 | -0.346 | **0.007** |
| **Step 2** |  |  |  |  |
|  | Total activity score | -0.006 | -0.315 | **0.012** |
|  | Age | 2.5 | 0.243 | 0.0503 |
|  | BMI | 2.7 | 0.160 | 0.191 |

$R^2$ at step 1 = .120, $R^2$ at step 2 = .199, $\Delta R^2$ = 0.08

by existing comorbidities and environmental factors, related to work and even seasons. Interestingly, after surgery, patients with lower activity scores appeared to have significantly higher peak levels of CRP, which may be a demonstration of the known anti-inflammatory benefits of exercise [6]. This increased inflammatory response to surgery was not, however, accompanied by postoperative complications or cognitive decline. Furthermore, there were no other differences in postoperative outcomes between inactive and active patients.

The proportion of patients classified as physically active in the current study was higher than anticipated. In fact, the levels of physical activity reported by our patients were similar to those found among the general population level of people 55 years and older in the Netherlands [55]. Previous literature from the Netherlands has indicated that the percentage of physically active patients undergoing cardiac surgery is between 56–57% [56–58]. Interestingly, the criteria used in those studies to determine preoperative levels of physical activity (the Corpus Christi Heart Project classification) was less demanding than the Dutch national recommendations, as it requires only 15 min/day of 'fairly light' intensity activities for a person to be considered physically active [59]. Furthermore, these studies were able to identify significant pre- and postoperative differences in outcome between active and inactive patients. For instance, Noyez et al. found that inactive patients were significantly more obese, older and had higher logistic Euroscores [57]. The high proportion of fit patients in the current study, may reflect the differences in patient population studied. The patients in the current study had a median (IQR) EuroSCOREs of 2.6 (1.5–4.2), which is associated with a low to medium cardiac risk, whereas the previous studies tended to include both older (mean (SD) age of 68.7 (10.9)) and higher-risk patients (a mean (SD) Logistic Euroscore of 5.06(5.6)) [56–58]. The unexpectedly high proportion of active patients in the present study may also have been the result of a selection bias (see below). Our results on preoperative physical activity, nevertheless indicate that there may be more variation in the larger cardiac surgery patient population than that which was previously assumed.

To the best of our knowledge, this is the first study to have used the SQUASH questionnaire to determine preoperative physical activity among cardiac patients. The questionnaire was developed to assess compliance of the Dutch population with the weekly duration and intensity of physical activity recommended by the RIVM. The SQUASH was found to be a reliable tool to assess physical activity behaviour in both the Dutch general population and Dutch patients who have undergone Total Hip Arthroplasty [33, 34]. In Denmark the SQUASH was found to be poor at measuring physical activity on an individual level but still reliable in distinguishing between individuals [60]. Nonetheless, the SQUASH remains the nationally standardised and recommended tool for assessment of physical activity in the Netherlands, and was selected for our study for this reason.

The SQUASH questionnaire, which asks participants to self-report their levels of physical activity and its intensity, has the same risks of biases suffered by many other questionnaires, as it provides subjective information. Patients were asked to report the intensity of their physical activity. The intensity of an activity experienced by a patient is a subjective construct. An activity that would normally be considered of moderate intensity to a healthy adult, could be considered vigorous by a cardiac patient, because they experience symptoms such as shortness of breath or chest pain during lower intensity activities. A customised questionnaire designed for a specific patient group would be necessary to account for this subjective construct. It is also possible that physically inactive patients may be somewhat embarrassed or ashamed by their lack of activity, or by guilt that their cardiac condition might have been caused by inactivity and thus self-inflicted. Despite the fact that their answers were treated anonymously, unfit patients may have been inclined to provide more socially acceptable answers, and thereby to have overestimated their overall level of fitness [33, 61, 62]. The SQUASH questionnaire may

offer an interesting tool to discriminate levels of physical activity when further compared to objective fitness assessments using activity monitors. Unlike activity monitors, the strength of using the SQUASH questionnaire is that it offers insight on the type and nature of physical activity that patients participate in. Furthermore, the SQUASH is both more feasible and generalizable to the clinical practice because it is brief and not as taxing as requiring patients to equip an activity monitor for multiple days.

Furthermore, many of our patients reported that they cycled frequently. Unfortunately, we did not ask them to report the type of bicycle they used, and electronic bicycles were starting to become more popular during the study. These bicycles offer varying degrees of assistance to the user, which makes it difficult to interpret the physical effort involved without further specification. Alternatively, electronic bicycles increases the duration of physical activity. Another likely source of bias is a degree of selection bias is that patients who are fitter and/or more self-aware of their health and activity may be likely to be more motivated to consent to involvement in a study of physical activity. Finally, although a degree of recall bias is also possible, some studies have demonstrated that retrospective and proxy reports of physical function do not undermine the predictive validity of questionnaires [63, 64]. Often where differences have been identified, they were limited primarily to mental health and disease symptoms rather than physical functioning [65, 66].

There is growing interest in the influence of frailty and sarcopenia on postoperative outcomes. One of the goals of our study was thus to explore the feasibility of assessing two components of frailty, weakness (handgrip strength) and slowness (get-up-and-go test) [41, 42], in the perioperative setting. Our study shows that these measurements are indeed feasible. In both active and inactive patients, handgrip strength was within the standard age and gender related ranges. With the exception of patients who underwent radial artery grafting, we were successful at measuring grip-strength bimanually four days after surgery. Preoperatively, the number of impaired hand grip strengths or get-up and go tests, indicative of frailty, were also similar between the active and inactive patients which suggests that few of our patients were markedly frail before surgery. Unlike the current study, previous literature has found that poorer preoperative grip strengths may be related to higher cardiac risk scores [67]. Handgrip strength also appears to be an effective tool at predicting postoperative complications after cardiac surgery [68]. Our study could have been enhanced by including CT scan measurements of total psoas area (a valid measure of sarcopenia and frailty), or of leg strength as an additional indicator of weakness. Had we included the latter, postoperative measurements would likely have been limited by leg pain related to saphenous vein harvesting. A larger, more adequately powered study would be necessary to confirm or refute the clinical usefulness of these measurements in a cardiac surgical population.

## Cognitive function

For the assessment of PNCD we applied the criterion recommended by the expert working party on post-operative neurocognitive disorders for major neurocognitive disorder–i.e. >2 standardised Z-score in cognitive function test scores [47]. As in previous studies on PNCD, we applied this stricter definition for PNCD and corrected for learning effects and age-related test-retest variability in neuropsychological test performance over time. The overall incidence of PNCD at 3-months after surgery was close to that which we anticipated, 26%, and is also consistent with incidences reported in earlier literature [69, 70]. The study was not powered to investigate this outcome, and so the following findings should be viewed in this light. No significant reduction of cognitive function was identified in patients in our study who were physically inactive compared with those who were active. Activity scores were also unrelated with

composite Z-scores. Although previous studies have associated grip strength with cognitive function, particularly of healthy older adults, our study could not confirm this relationship. Pre-existing medical conditions, as well as postoperative complications did not appear to be associated with PNCD or composite Z-score.

The beneficial effects of physical activity on the brain and on cognitive function have been well documented in human and animal studies. [6, 14–17] Adequate levels of preoperative physical activity may improve cognitive function and thus the preoperative cognitive reserve mentioned above, and may thereby help reduce the risk of postoperative cognitive decline. Indeed, Hayashi et al. reported that a lower preoperative exercise capacity, assessed by the 6-minute-walking distance in patients undergoing cardiac surgery was associated with an increased incidence of PNCD [71]. However, their study assessed cognitive function only 14 days after surgery using the mini mental state examination, which is a sub-optimal tool for detecting PNCD due to both floor- and ceiling-effects, and poor sensitivity for detecting mild or medium cognitive impairment [72, 73]. Furthermore, a decline in cognitive performance at 14 days after surgery is not necessarily caused by persistent neurocognitive disorder, as this is also likely to be related to delirium or delayed cognitive recovery caused by pain, residual drug effects, limited mobility, and fatigue [46, 47]. Contrary to Hayashi et al.'s findings, we found no association between preoperative activity scores, muscle strength and mobility, and a reduced cognitive performance or PNCD.

Our study has some limitations, some of which have already been highlighted and addressed. The primary aim of this study was to investigate the level of fitness of patients undergoing cardiac surgery. For this reason, a pragmatic sample size of 100 patients was selected. In an attempt to include a somewhat homogenous population, we included elective CABG surgery patients, and excluded patients undergoing combined CABG and valve replacement procedures. The patients included in our study had a low risk of postoperative complications. The small number of physically inactive patients may have been due to selection and inclusion biases, and the low incidence of postoperative complications might have been influenced by the low-to-medium risk levels of the patients (as suggested by their EuroSCORE). Furthermore, as inflammation was only a secondary parameter in this study, data regarding preoperative medications was not collected. Certain medications are known to effect pre- and postoperative inflammation. Therefore, we recommended that future studies assessing peri- and postoperative inflammation should include medication history within their analyses.

During this study preoperative physical activity was measured once, at the preoperative screening. The waiting time between screening and surgery was >4 weeks for more than half of our patients. During this period some patients may have improved or lowered their activity levels, unknown to us. Although we were able to determine the level of physical activity of patients who need to undergo cardiac surgery, future research on this topic should assess physical function closer to the surgery date in order to gain a more accurate indication of the level physical activity for patients about to enter cardiac surgery.

More recently the WHO has placed more emphasis on strength training. After our study was finished, the RIVM adjusted their recommendations for physical activity to also put emphasis on muscle strengthening activities and reduced sedentary lifestyles [37, 74]. In the Netherlands, this new guideline (the so-called 'Beweegrichtlijn') has adjusted the MET values for certain activities and in addition they now recommend that adults also do muscle and skeletal-strengthening activities at least twice a week [37]. As we did not include questions specifically directed towards strength training in our study, we cannot retrospectively assess compliance with these new guidelines. Among the general population it is known that only 45% meet the recommendations of the newer guideline, whereas 75% of the general adult

population met the Combinorm criteria that were used in our study [55]. Our expectation is that if we were able to apply the newer RIVM guidelines, we would have found a lower proportion of patients meeting the criteria for healthy levels of physical activity. New research is needed to assess compliance among cardiac (surgery) patients with these newer guidelines.

In summary, we are the first to have used the SQUASH questionnaire to quantify levels of physical activity among cardiac surgical patients. We have found that among 100 low to medium-risk patients undergoing elective coronary artery bypass surgery, 76% fulfilled the then current Dutch guidelines for physical activity. Although a significant association was found between preoperative levels of physical activity and cardiac risk (indicated by logistic EuroSCORE), studies are needed to determine if this relationship is causal. Consistent with previous studies, the degree of physical activity appeared to correlate significantly with the level of inflammatory response to surgery, as measured in peak postoperative CRP. This however, was not associated with postoperative complications or cognitive performance. This finding requires further verification in studies specifically powered to examine these associations. Finally, the SQUASH, as well as measurements for grip strength were easy to administer and feasible to use in our population of patients.

## Supporting information

**S1 Table. Components of the Short Questionnaire to Assess Health (SQUASH).**
(DOCX)

**S2 Table. Different intensity categories for physical activities based on age and Metabolic Equivalent of Task (MET) according to values assigned in the Ainsworth compendium for physical activity.**
(DOCX)

**S3 Table. Univariate logistic regression for patients meeting the preoperative physical activity according to the combinorm.**
(DOCX)

**S4 Table. Pre-operative physical activity and frailty assessments.**
(DOCX)

**S5 Table. Univariate and multivariate logistic regression analyses for Postoperative Neurocognitive Disorder (PNCD).**
(DOCX)

**S6 Table. Standardized scores for each cognitive domain prior to surgery, physically active vs. physically inactive.**
(DOCX)

**S7 Table. Preoperative neuropsychological and wellbeing test results.**
(DOCX)

## Acknowledgments

We would like to thank emeritus professor Luc van der Woude for his insightful comments and suggestion throughout the writing of this article.

## Author Contributions

**Conceptualization:** Sawal D. Atmosoerodjo, Iris Tigchelaar, Rolf Huet, Massimo A. Mariani, Anthony R. Absalom.

**Data curation:** Setayesh R. Tasbihgou, Sandra Dijkstra.

**Formal analysis:** Setayesh R. Tasbihgou.

**Investigation:** Setayesh R. Tasbihgou, Sandra Dijkstra, Iris Tigchelaar, Rolf Huet.

**Methodology:** Sawal D. Atmosoerodjo, Rolf Huet, Massimo A. Mariani, Anthony R. Absalom.

**Project administration:** Rolf Huet, Anthony R. Absalom.

**Resources:** Anthony R. Absalom.

**Supervision:** Rolf Huet, Anthony R. Absalom.

**Writing – original draft:** Setayesh R. Tasbihgou.

**Writing – review & editing:** Setayesh R. Tasbihgou, Sandra Dijkstra, Sawal D. Atmosoerodjo, Iris Tigchelaar, Rolf Huet, Anthony R. Absalom.

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
