## [Decision Letter · Decision Letter 0]

24 Aug 2020

PONE-D-20-23287

A prospective pilot study assessing levels of preoperative physical activity and postoperative neurocognitive disorder among patients undergoing elective coronary artery bypass graft surgery

PLOS ONE

Dear Dr. Tasbihgou,

Thank you for submitting your manuscript to PLOS ONE. After careful consideration, we feel that it has merit but does not fully meet PLOS ONE’s publication criteria as it currently stands. Therefore, we invite you to submit a revised version of the manuscript that addresses the points raised during the review process.

In order to expedite the review of this paper, an academic editor along with 1 reviewer has read and reviewed your paper. As academic editor, I found the study to be an interesting one. I believe that pre and post cardiac rehab along with improved overall physical conditioning are very

important in determining postoperative outcome. As a reviewer, I have also provided some points of thought for the authors. 

We look forward to receiving your revised manuscript.

Kind regards,

Salil Deo

Academic Editor

PLOS ONE

Additional Editor Comments:

Study design: Prospective study

Evaluation: Preoperative evaluation of physical activity, handgrip strength and chair test to assess for frailty, collection of routine demographic and other lab/ clinical details regarding patients.

Sample size: 100 based on power calculations.

Comments:

1.Inclusion criteria - Patients were required to be able to stand and walk independently --- does this mean that those using a walker or cane were excluded from the study ? These patients are frail, yet with the changing demographics of patient population, a larger % of such patients are being referred for surgery. Same goes for impaired hearing/eyesight --- we need to understand what degree of impairment was used as a cut-off. From sample size calculations, authors want to include 100 patients. However due to starting the prehab program, they were limited to 62 patients. Can authors please explain how they are expected to fulfil power for hypothesis testing after losing 31 patients ?

2.The paragraph on surgery and anesthesia can be removed. It does not contribute to the understanding of results. A simple statement “ Surgery was performed in the conventional manner with routine pre and postoperative care “ can suffice here.

3.I would recommend authors to combine tables 1/2/3 into a more concise format to present only that information that is being used to compare and report results in these patients. More details regarding the scale and components of the scale can be presented in the supplement, which interested readers can then see. Reducing these tables would help to improve the flow of the paper and prevent reader from being distracted by many different parameters. It appears that the authors are finally using the total activity score.

4.Table 5 --- Please present some values for hypothesis test results either conventional p-values or standardized differences, so that readers can get an understanding of how different these groups are.

5.Please allow the paragraphs to flow and include figure explanations inside the paragraph, rather than a separate sentence as done right now.

6.It would be important to know if any of these patients had recent NSTEMI prior to surgery. That may increase CRP levels.

7.Were any of the patients tested at 4 weeks part of a post-operative cardiac rehab program ?

8.Do authors have data on preoperative medications that may influence inflammation ? Statins and other medications on the basis of pleiotropic effect may change CRP levels.

Authors have conducted an interesting study; it is likely that the study suffers from low sample size and hence cannot identify a difference in end-points between physically active and inactive. If authors have data regarding frail patients or those that on the basis of their criteria are too physically deconditioned to be a part of the study, I am thinking that such a group may serve as a good control to contrast the results that the authors found in this study. This is just a suggestion; not a recommendation.

Journal Requirements:

"SRT, SWD, IT and RH have declared that no competing interests exist.

SD has read the journal's policy and the authors of this manuscript have the following competing interests: grants from Stichting Beatrixoord Noord-Nederland during the conduct of the study. This grant is not related in anyway to the study;

MAM has read the journal's policy and the authors of this manuscript have the following competing interests:consultancy from AtriCure, Getinge and LivaNova;

ARA has read the journal's policy and the authors of this manuscript have the following competing interests: reports unrestricted research and/or consultancy for The Medicines Company, Janssen Pharma, Carefusion/BD, Orion Pharma, Ever Pharma, and Philips (all for work unrelated to the current study; all payments to institution);  and being Editor of the British Journal of Anaesthesia"

Reviewers' comments:

Reviewer's Responses to Questions

**Comments to the Author**

1. Is the manuscript technically sound, and do the data support the conclusions?

Reviewer #1: Yes

2. Has the statistical analysis been performed appropriately and rigorously? 

Reviewer #1: Yes

3. Have the authors made all data underlying the findings in their manuscript fully available?

Reviewer #1: Yes

4. Is the manuscript presented in an intelligible fashion and written in standard English?

Reviewer #1: Yes

5. Review Comments to the Author

Reviewer #1: Dear the authors of the manuscript entitled " A prospective pilot study assessing levels of preoperative physical activity and postoperative neurocognitive disorder among patients undergoing elective coronary artery bypass graft surgery"

Thank you for writing this manuscript which describes effect of physical activity assessed by SQUASH questionnaire on cognitive performance and other post operative complications in low to moderate risk patients who underwent coronary artery bypass surgery.

The study demonstrated that physical activity did not show major impact on post operative outcomes, and to my knowledge is the first to handle this subject using SQUASH questionnaire for such assessment.

Study conduct, data analysis and literature review were optimum

English language was competent

I was glad reading this manuscript and have no concerns about it

Thank you

6. PLOS authors have the option to publish the peer review history of their article (what does this mean?). If published, this will include your full peer review and any attached files.

Reviewer #1: **Yes: **Salah Altarabsheh

---

## [Author Response · Author response to Decision Letter 0]

30 Aug 2020

Dear Dr. Salil Deo,

We would like to thank you and the reviewer for taking the time to review our manuscript. We appreciate the helpful remarks and have made every effort to revise the manuscript.

Editor Point 1a

“Inclusion criteria - Patients were required to be able to stand and walk independently --- does this mean that those using a walker or cane were excluded from the study? These patients are frail, yet with the changing demographics of patient population, a larger % of such patients are being referred for surgery.”

Response:

We thank the editor for his/her thorough reading of our manuscript and the suggestions for improvement. Indeed, we required patients to be able to walk independently. This was to enable an objective measurement of performance in the get-up-and-go test. None of the included patients used a cane or a walker. For the purposes of the study we would have regarded patients as unsuitable if they were unable to walk without a cane or walker or without support from another person. 

Editor Point 1b

“Same goes for impaired hearing/eyesight --- we need to understand what degree of impairment was used as a cut-off.” 

Response:

We have looked again at the wording of our ethical committee approved protocol and study notes. The wording we used in our manuscript was a précis of the exclusion criteria that we applied. The exact inclusion and exclusion criteria applied were (copy/paste from our IRB approved protocol): 

“In order to be eligible to participate in this study, a subject must meet all of the following criteria: 

• Scheduled for elective cardiac coronary surgery, and booked for routine clinical assessment on the cardiosurgical preoperative screening unit.

• Able to stand and walk independently

• Able to participate in the online screening modules for cognitive function (ie able to operate a computer touch pad or mouse, and to read large text on a computer screen).

• They should be prepared to allow a researcher to visit them at home 3 months after their operation.

• Patients need to be able to perform the handgrip strength test on both sides.

A potential subject who meets any of the following criteria will be excluded from participation in this study: 

• Extended postoperative ICU stay is expected.

• Inability to understand or read Dutch instructions

• Recent history of depression or severe anxiety

• History of dementia or other neurological disorders

• History of stroke, or other severe cerebrovascular insults

• Patient is not able to perform get-up-and-go test or any of the other tests”

We regret the vague and inaccurate wording of the exclusion criteria in our initial submission, and for the sake of better transparency and accuracy, we have revised the text in the manuscript to include copies of the text for the above criteria. 

To further answer your question, we mention the following. Formal hearing and vision tests were not done and so no specific cut-off was used to determine these impairments. Rather, the screening investigator made a pragmatic judgement. If patients were suffering from a hearing/eyesight impairment to an extent which would prevent them from following instructions or complete any of the assessments, they were deemed unsuitable and were also excluded from the study. Put another way, patients were accepted if was acceptable if they had with hearing/ eyesight impairment that did not prevent them following being able to perform the study procedures. 

Editor Point 1c

“From sample size calculations, authors want to include 100 patients. However due to starting the prehab program, they were limited to 62 patients. Can authors please explain how they are expected to fulfil power for hypothesis testing after losing 31 patients?”

Response:

With regard to the sample size analysis; the main goal of our research was to evaluate the degree of preoperative physical activity of our hospital’s cardiac surgery patient population, prior to deciding whether or not a prehabilitation program was warranted. As such no specific hypotheses were being tested and no sample size analysis was conducted. When designing the study there was no data available to inform a sample size calculation, therefore the sample size of 100 patients was a pragmatic choice, which we believed would provide an estimate of the physical activity levels of our patients, with a reasonably narrow confidence interval. The results of the current study would then provide a basis for future studies specifically assessing the relationship between preoperative physical activity and postoperative outcome. For this, our primary goal, we were able to use the data from 100 patients, as planned.

The analyses of relationships among physical activity and secondary outcome variables, were secondary study goals. The power analysis described in the first version of our manuscript was done in order to give an impression of what proportion of physically active and inactive patients would be necessary to identify relationships among pre-operative physical activity and secondary endpoints. As it turned out, following pressure of increased clinical insight, our surgical colleagues adapted their clinical preoperative practice and sent 31 patients to a preliminary prehabilitation program during the execution of our study. We decided not to use the data for these 31 patients for our secondary analyses. 

On reflection, the text about a power analysis was confusing anyway, because our sample size was pragmatically chosen as described above and in the manuscript. To avoid confusion, we have deleted the text regarding the power analysis for the secondary analysis (which was meant as hypothesis forming for future studies). 

Editor Point 2

“The paragraph on surgery and anesthesia can be removed. It does not contribute to the understanding of results. A simple statement “Surgery was performed in the conventional manner with routine pre and postoperative care” can suffice here.”

Response:

We agree and have replaced the original paragraph in the manuscript with a statement.

Editor Point 3

“I would recommend authors to combine tables 1/2/3 into a more concise format to present only that information that is being used to compare and report results in these patients. More details regarding the scale and components of the scale can be presented in the supplement, which interested readers can then see. Reducing these tables would help to improve the flow of the paper and prevent reader from being distracted by many different parameters. It appears that the authors are finally using the total activity score.”

Response:

We agree and have removed these tables from our manuscript and included them into the supplement. Indeed, when analyzing postoperative data, we used the (continuous) total activity score to differentiate the varying degrees of physical activity between our patients. However, the (binary) general recommendations for physical activity (combinorm), provided by the Dutch National Institute for Public Health and Environment (Rijksinstituut voor Volksgezondheid en Milieu, RIVM), were used to analyze preoperative levels of physical fitness.

Editor Point 4

“Table 5 --- Please present some values for hypothesis test results either conventional p-values or standardized differences, so that readers can get an understanding of how different these groups are.”

Response:

Thank you for this suggestion. We have incorporated p-values into the table. 

Editor Point 5

“Please allow the paragraphs to flow and include figure explanations inside the paragraph, rather than a separate sentence as done right now”

Response:

In our first submission, the text of the different paragraphs includes references to the figures in the usual way, with an explanation where necessary. In accordance with our understanding of the author instructions, we placed the legend for each figure after the paragraph first referencing it. We apologize if we misunderstood the author guidelines. Please let us know if you want us to move the figure legends to a separate page at the end of the document. 

Editor Point 6

“It would be important to know if any of these patients had recent NSTEMI prior to surgery. That may increase CRP levels.”

Response:

In our institution, the preferred choice of procedure for a STEMI or non-STEMI is coronary angiography and percutaneous coronary intervention. Depending on the severity and the accompanied calculated risk score, the treatment for a non-STEMI may be done urgently, semi-urgently or even electively. Our study aimed to include patients undergoing elective CABG surgery. 

In response to your question, we have looked again closely at our data. Among the 100 patients included into our study, 6 were known to have had a NSTEMI in the years preceding the CABG, of who only 3 patients had had a myocardial infarction within 1 month prior to surgery. Of these 3, 2 were physically active and 1 physically inactive according to the Dutch guidelines. The preoperative CRP levels in these 3 patients were not markedly increased (they were: 13, 18 and 23). The results of our study do not appear to be strongly affected by these patients. We have repeated our analyses without the data from these patients – in this new analysis the median (IQR) level of preoperative CRP is 1.6 (0.7-3.1) and 3.5 (1.4-12) for physically active and inactive patients respectively (originally 1.8 (0.7-3.3) and 3.6 (1.5-12)). Furthermore, the difference between the active and inactive groups is unchanged and remains statistically significant (p=0.006). For now, we have not changed the text of our manuscript in this regard, but would be happy to do so, if you want us to. 

Editor Point 7

“Were any of the patients tested at 4 weeks part of a post-operative cardiac rehab program?”

Response:

We are uncertain what you are asking. We performed hand grip strength tests at 4 days as reported in the results section of the paper. The results presented are of patients who did not undergo the preoperative and postoperative rehabilitation program. Our study protocol did not involve tests at 4 weeks post-operatively. 

Patients who were included by the surgeons in the postoperative rehabilitation program underwent a program of aerobic cycling, resistance training, swimming, and sport and games, but no tests, as this program was incorporated into clinical care and not part of a study. 

Editor Point 8a

“Do authors have data on preoperative medications that may influence inflammation? Statins and other medications on the basis of pleiotropic effect may change CRP levels.”

Response:

Regrettably, we did not record this information during the execution of the study, as inflammation was a secondary parameter. We agree that information on preoperative medication would be relevant for the analyses of peri- and postoperative inflammation, and have added a sentence to the limitations section about this.

Editor Point 8b

“Authors have conducted an interesting study; it is likely that the study suffers from low sample size and hence cannot identify a difference in end-points between physically active and inactive. If authors have data regarding frail patients or those that on the basis of their criteria are too physically deconditioned to be a part of the study, I am thinking that such a group may serve as a good control to contrast the results that the authors found in this study. This is just a suggestion; not a recommendation.”

Response:

Thank you for this suggestion. Indeed, data regarding frail patients would provide additional insight on the role of preoperative physical condition in surgery and postoperative outcome. Unfortunately, we do not have such data and so cannot perform the suggested secondary analysis. 

We trust that these responses and revisions meet with your satisfaction. 

Kind regards,

SR Tasbihgou

---

## [Editor Report · Decision Letter 1]

21 Sep 2020

A prospective pilot study assessing levels of preoperative physical activity and postoperative neurocognitive disorder among patients undergoing elective coronary artery bypass graft surgery

PONE-D-20-23287R1

Dear Dr. Tasbihgou,

We’re pleased to inform you that your manuscript has been judged scientifically suitable for publication and will be formally accepted for publication once it meets all outstanding technical requirements.

Kind regards,

Salil Deo

Academic Editor

PLOS ONE

Additional Editor Comments (optional):

All comments have been answered and required modifications have been addressed.
---

## [Editor Report · Acceptance letter]

28 Sep 2020

PONE-D-20-23287R1 

A prospective pilot study assessing levels of preoperative physical activity and postoperative neurocognitive disorder among patients undergoing elective coronary artery bypass graft surgery 

Dear Dr. Tasbihgou:

I'm pleased to inform you that your manuscript has been deemed suitable for publication in PLOS ONE. Congratulations! Your manuscript is now with our production department. 

Kind regards, 

on behalf of

Dr. Salil Deo 

Academic Editor

PLOS ONE